# Indoor Formaldehyde Concentration, Personal Formaldehyde Exposure and Clinical Symptoms during Anatomy Dissection Sessions, University of Medicine 1, Yangon

**DOI:** 10.3390/ijerph18020712

**Published:** 2021-01-15

**Authors:** Win-Yu Aung, Hironari Sakamoto, Ayana Sato, Ei-Ei-Pan-Nu Yi, Zaw-Lin Thein, Myint-San Nwe, Nanda Shein, Htin Linn, Shigehisa Uchiyama, Naoki Kunugita, Tin-Tin Win-Shwe, Ohn Mar

**Affiliations:** 1Department of Physiology, University of Medicine 1, Yangon 11014, Myanmar; wyuaung@gmail.com (W.-Y.A.); dreepny@gmail.com (E.-E.-P.-N.Y.); dr.zawlinthein@gmail.com (Z.-L.T.); dr.ohnma@gmail.com (O.M.); 2Faculty and Graduate School of Engineering, Chiba University, Chiba 263-8522, Japan; hironari-sakamoto@chiba-u.jp (H.S.); ayana.sato08@gmail.com (A.S.); uchiyama.s.aa@niph.go.jp (S.U.); 3Department of Anatomy, University of Medicine 1, Yangon 11014, Myanmar; dr.myintsannwe1960@gmail.com (M.-S.N.); 2007.nanda@gmail.com (N.S.); htinlinn2008@gmail.com (H.L.); 4Department of Environmental Health, National Institute of Public Health, Saitama 351-0197, Japan; 5School of Health Sciences, University of Occupational and Environmental Health, Kitakyushu 807-8555, Japan; kunugita@med.uoeh-u.ac.jp; 6Center for Health and Environmental Risk Research, National Institute for Environmental Studies, Tsukuba-City 305-8506, Japan

**Keywords:** formaldehyde, indoor concentration, personal exposure, clinical symptoms, anatomy dissection

## Abstract

The formaldehyde (FA) embalming method, the world’s most common protocol for the fixation of cadavers, has been consistently used in medical universities in Myanmar. This study was designed to examine the indoor FA concentrations in anatomy dissection rooms, an exposed site, and lecture theater, an unexposed control site, and to access personal exposure levels of FA and clinical symptoms of medical students and instructors. In total, 208 second year medical students (1/2019 batch) and 18 instructors from Department of Anatomy, University of Medicine 1, participated. Thirteen dissection sessions were investigated from February 2019 to January 2020. Diffusive sampling devices were used as air samplers and high-performance liquid chromatography was used for measurement of FA. Average indoor FA concentration of four dissection rooms was 0.43 (0.09–1.22) ppm and all dissection rooms showed indoor concentrations above the occupational exposure limits and short-term exposure limit for general population. Personal FA exposure values were higher than indoor FA concentrations and the instructors (0.68, 0.04–2.11 ppm) had higher exposure than the students (0.44, 0.06–1.72 ppm). Unpleasant odor, eye and nose irritations and inability to concentrate were frequently reported FA-related symptoms, and the students were found to have significantly higher risks (*p* < 0.05) of having these symptoms during the dissection sessions than during lecture.

## 1. Introduction

For hundreds of years, human anatomy has been a keystone of the medical education. An understanding of structures of the human body provides a platform of knowledge suitable for all medical professionals [1]. For learning human anatomy, an important learning resource is a cadaveric dissection, a cornerstone in the training of medical students for the development of 3D awareness of the human body in both visual and tactile ways [2]. To preserve cadavers, embalming is mandatory to get tissue preservation with minimal structural changes [3].

A common purpose of all embalming protocols is the use of biocidal substances having effectiveness in prevention of tissue decay and killing of pathogens. An accumulating body of literature indicates that formaldehyde (FA) embalming method can still be considered the world’s most common protocol for the fixation of cadavers [4], although there are some additional procedures such as Thiel’s embalming method [5], ethanol–glycerin embalming [6] and nitrite pickling salt embalming [7].

Exposure to FA occurs primarily by inhalation, and various acute and chronic health effects are reported. Many animal studies showed that FA inhalation exposure could lead to dyspnea, vomiting, hypersalivation, muscle spasms and death [8]; severe irritation and damage to the epithelium of the nasal cavity [9]; neurobehavioral effects such as altered learning and memory, decrease in exploratory behaviors and spontaneous motor activity [10]; and neural and immune dysfunctions [11]. Likewise, human studies have shown that FA exposure is associated with irritation of eyes, nose and throat and respiratory problems such as bronchoconstriction and pneumonia [12]. Being an effective biocidal substance with germicidal, bactericidal and fungicidal activities, FA, at the same time, is toxic and can be mutagenic and carcinogenic. Based on the evidence of the relationship between nasopharyngeal cancer and FA exposure, the International Agency for Research on Cancer classified FA as a human carcinogen (Group 1A) [13]. Moreover, FA could probably cause the mutation required for the development of leukemia [14].

With reports of such various health hazards association with FA exposure, there has been an emergence of deep concern on indoor air quality of anatomy dissection rooms/laboratories in term of FA and exposure of medical students and instructors to FA during dissection sessions. Several previous studies have reported that indoor FA concentrations of dissection rooms exceeded the recommended limits—0.23–1.08 ppm in Japan [15,16], 0.12–9.16 ppm in Korea [17], 0.4–0.5 ppm in Singapore [18] and 0.52–1.48 ppm in USA [19]—and relatively few studies reported on personal FA exposure with the findings of higher personal exposure than the indoor concentration [20,21,22].

In Myanmar, according to the curriculum of medical universities, human cadaveric dissection is the main part of the curriculum in the study of human anatomy. Instructors and second year medical students routinely handle the cadavers embalmed with formalin resulting in an inevitable consistent FA exposure in the dissection rooms for an entire academic year, a considerable duration. Even though traditional FA embalming method has been consistently used in Myanmar medical universities, there is no previous investigation on indoor FA concentration and personal FA exposure in the anatomy dissection rooms.

In addition, according to a review of literature, some factors such as inadequate ventilation system [23], increase in number of cadavers [24] and numbers of students per cadaver could lead to high FA exposure. Currently, in the dissection rooms of the Department of Anatomy, University of Medicine 1, Yangon, Myanmar, only wall fans are used for forced ventilation, 2–3 cadavers are present in one dissection room and 25–30 students are allotted to one cadaver. Under such circumstances, it is now critical to explore whether Myanmar medical students and instructors encounter high FA exposure or not. Therefore, in this study, we aimed to investigate indoor concentrations and personal exposure levels of FA in the dissection rooms as well as assess any relationship of FA exposure to clinical symptoms.

## 2. Materials and Methods

### 2.1. Subjects

Second-year medical students (1/2019 batch) and the instructors from the Department of Anatomy, UM1, were requested for voluntary participation to our study. They were initially explained the aim and the detailed procedure and informed on the exclusion criteria such as history of asthma, smoking habits and frequent use of body spray or perfume. Then, they were invited to take part in the study voluntarily. After receiving agreement from their parents/guardians and assent from them, the students were randomly selected to assign into exposed group and non-exposed group.

During the study period, from February 2019 to January 2020, 208 medical students were eligible, of whom 104 students (65% male) were assigned to the exposed group and 104 students (63% male) were assigned to the unexposed group. Meanwhile, the instructors took part repeatedly for personal FA exposure assessment for the reason that only 18 teaching staff members (16% male) were available during the study period.

All participants (students and instructors) were given informed consent in native Myanmar language to sign before enrolling in this study. They had the right to withdraw their consent at any time. This study was approved by the Research and Ethics Committee, UM1, Yangon (003/UM1, REC.2019) and also was conducted in accordance with the declaration of Helsinki 1975 (revised version 2013).

### 2.2. Study Location

The study locations were gross anatomy dissection rooms at the Department of Anatomy (exposed site) and lecture theater (LT) (unexposed site), Pyay campus, UM1, Yangon. There are five dissection rooms (DRs), but, during the study period, only four rooms (DR I, II, III and V) were used for the demonstration of gross anatomy dissection. Each room was provided with 10 wall fans, which were opened during the dissection session. Two entrance doors are present in each room, which remain closed when not in use. There are three fixed dissection tables in DR I and DR II and two tables in DR III and DR V. The description and characteristics of the study sites are shown in Figure 1 and Table 1, respectively.

### 2.3. Preparation of Formaldehyde-Embalmed Cadaver

The donated cadavers were perfused via femoral arteries with the embalming solution consisting of 40% FA (2 L), methylated spirit (1 L) and glycerol (500 mL) diluted with water to the final volume of 8–10 L depending on the size and body weight of the cadavers. Therefore, the final concentration of FA of the perfused embalming solution is reduced to 6–8% FA. The cadavers were then stored in the cadaver storage room which was separately situated from the dissection rooms (Figure 1). The cadaver storage tanks are filled with the embalming solution containing 4% FA. During the dissection session, each cadaver was allotted to 25–30 medical students and an instructor.

### 2.4. Sampling Procedures

The sampling procedures were performed by the investigators from the Department of Physiology and Department of Anatomy, UM1. The medical students were divided into two groups Groups A and B. According to the teaching schedule, while one group was assigned to the dissection session, another group had to attend the lecture. During the study period, neither Group A nor Group B was consistently the exposed group but could be the exposed group alternatively. Therefore, we were able to perform the sampling procedure in the dissection rooms and the lecture theater almost simultaneously. During the study period, 13 dissection sessions were investigated, and the durations of the dissection sessions varied from 1 to 2 h.

DSD-DNPH diffusive samplers [25] provided by National Institute of Public Health, Wako-shi, Saitama, Japan were used to assess indoor concentrations and personal exposure of FA. The samplers are small, lightweight and do not require a power source. They are stable for one month after collection [26] and enable the estimation of the time-weighted concentration of carbonyl compounds including FA [25]. The sampling rate is fairly constant between 30 and 240 min and average sampling rate is 71.9 mL/min. Therefore, the sampling rate and capacity of the sampler are not time-dependent [25].

Approximately at the center of each dissection room, an air sampler was adhered 6 feet above the floor (Figure 2A). In the lecture theater, there are 15 rows and an air sampler was set in the 3rd row and another one in the 12th row hanging one foot above the average height of the students. The positions of the air samplers did not hinder the activities of the students. Six diffusive air samplers were utilized for one cross-sectional measurement of indoor FA concentrations. Data loggers for temperature and relative humidity were placed close to the samplers. In addition, to compare with WHO recommended short term exposure limit of 30 min, we also performed 30 min air sampling in the dissection rooms (12 DSD-DNPH diffusive samplers were used).

For inhalational personal FA exposure level assessment, the DSD-DNPH samplers placed in plastic tubes with small pores were attached to collars of white coats (breathing zone) (Figure 2B). For each dissection session, eight students (3 in DR I, 3 in DR II, 1 in DR III and 1 in DR V) and six instructors (2 in DR I, 2 in DR II, 1 in DR III and 1 in DR V) were randomly selected to have personal samplers attached until the dissection session was finished. At the same time, eight students and a lecturer in the lecture theater were asked to have personal samplers attached to collars of shirts or blouses until the lecture was completed.

At the end of the dissection session and lecture, both indoor air samplers and personal samplers were collected, packed in respective packs with code numbers and stored at −20 °C at the Postgraduate Research laboratory, Department of Physiology. Then, the samplers were sent with a proper cold chain to the Laboratory of NIPH, Japan, for measurement of FA concentration by high-performance liquid chromatography (HPLC) method.

### 2.5. Air Sample Analysis

The detailed procedure was described in our previous study [27]. The HPLC system (Prominence LC-20, Shimadzu, Kyoto, Japan) was used with two LC-20AD pumps, an SIL-20AC autosampler and an SPD M20A photo-diode array detector. The analytical column was an Ascentis RP-Amide, 3 µm particle size, 150 mm × 3 mm i.d. column (Milipore Sigma, Bellefonte, PA, USA). Solution A of the mobile phase mixture was acetonitrile/water (50/50 *v*/*v*) containing 10 mmol/L ammonium acetate and Solution B was acetonitrile/water (80/20 *v*/*v*). HPLC elution was carried out with 100% A for 5 min, followed by a linear gradient from 100% A to 100% B in 50 min and then held for 10 min. The flow rate of the mobile phase was 0.8 mL/min. The column temperature was 35 °C and the injection volume was 10 µL.

### 2.6. Questionnaire for Assessment of Formaldehyde-Related Clinical Symptoms

For the assessment of FA-related symptoms experienced during acute exposure, 104 medical students were randomly selected by lottery method after excluding those with a history of allergy or skin disorders. A modified form of checklists of symptoms questionnaires from Formaldehyde Medical Disease Questionnaire of Occupational Safety and Health Assessment was used to assess FA-related symptoms including unpleasant smell, eye itchiness, eye redness, excessive lacrimation, blurred vision, nasal symptoms (e.g., nasal congestion, dryness and soreness), dry mouth, headache, nausea and vomiting, cough, wheezing, shortness of breath, general fatigue, dizziness, itchiness of skin and inability to concentrate. The response was Yes or No to the symptoms regarding two different environments—in the dissection rooms and in the lecture theater (Figure 3). The students were asked to give the responses by recalling that the listed symptoms were felt or not in these two different conditions. No symptom was reported in both two conditions.

This assessment was done near the end of the study period to avoid the effects of fluctuations in indoor FA concentrations between the dissection sessions. In total, 104 students returned the questionnaire and all responders had already experienced acute FA exposure during their respective dissection sessions; some of them were once the subjects having personal samplers attached to the breathing zone during the dissection sessions and some were during lecture periods. Moreover, we also asked them about the regular use of personal protective equipment such as laboratory coat, surgical gloves, surgical masks and goggles during the dissection sessions.

### 2.7. Statistical Analysis

Statistical Package for the Social Sciences (SPSS) version 26 (IBM Corp., Armonk, NY, USA) was used for data entry and statistical analysis. After data entry was completed, data cleaning was done by checking with descriptive statistics including number, minimum, maximum, mean, standard deviation (SD), median, interquartile range (IQR), standard error of mean (SEM), frequency distribution and box plot figures.

Statistically significant level was determined at *p* < 0.05. For comparison of indoor FA concentrations among the dissection rooms and the lecture theater, one-way ANOVA with post-hoc Bonferroni correction was used. Personal FA exposure levels between exposed instructors and exposed students in each dissection room were compared and analyzed by independent sample t test. A relationship between FA exposure and clinical symptoms was analyzed by simple logistic regression from paired samples and expressed as odd ratio (OR) with 95% confidence interval (CI).

## 3. Results and Discussion

### 3.1. Indoor Formaldehyde Concentrations

Average indoor concentrations of FA in four DRs and LT are shown in Figure 4. Indoor FA concentrations in all dissection rooms were above the occupational exposure limit-ceiling (OEL-C), 0.1 ppm, set by the Japan Society for Occupational Health [28]. Moreover, those of DR I, DR II and DR III also exceeded occupational threshold time value-ceiling (TVL-C), 0.3 ppm, established by the American Conference of Governmental Industrial Hygienists [29]. In the lecture theater, an unexposed site, the average indoor FA concentration was not only below the guideline values but also was significantly lower than that of dissection rooms (*p* = 0.008). This result indicates that the indoor air quality in the dissection rooms in terms of FA concentration was unsatisfactory.

Although DR I vs. DR II and DR III vs. DR V have the same dimensions and the same number of cadavers, respectively (Table 1), two similar rooms still showed different indoor FA concentrations (Figure 4). DR I showed the highest indoor FA concentration with statistically significant difference from the remaining three rooms (*p* = 0.041 between DR I and DR II, *p* = 0.035 between DR I and DR III and *p* = 0.014 between DR I and DR V) but no statistically significant difference was seen among DR II, DR III and DR V. Differences in body size, sex and amount of subcutaneous adipose tissues of dissected cadavers may be a possible explanation, but such detailed characteristics of the cadavers were not recorded in our study. A previous study gave an account that female cadavers released higher levels of FA than male cadavers and suggested that subcutaneous adipose tissues could be one of the emitting sources of FA [30].

The most common short term exposure limit (STEL) for FA is 0.08 ppm for a 30 min average value recommended by the World Health Organization aiming at preventing significant sensory irritation in the general population [12]. The Ministry of Health, Labor and Welfare of Japan also established the upper limit of FA in domestic rooms as 0.08 ppm for any 30 min duration [31]. In our study, we also carried out short-term exposure limit sampling for 30 min in the dissection rooms, and it was noticed that each dissection room showed higher indoor FA concentrations than these STEL values (Figure 5). This finding indicates that, during dissection sessions, all subjects had higher FA exposure than the general population.

As indicated by the comparisons with recommended values, we observed that indoor FA concentrations of the dissection rooms in our study were above these values in terms of either short-term exposure limit value for general population [12,31] or occupational limit values [28,29]. Several previous studies, from Singapore [18], Korea [17], USA [19], Japan [15,16], Thailand [32], Saudi Arabia [20], Brazil [22] and India [33], also reported that indoor FA concentrations in anatomy dissection laboratories were above the recommended levels of respective international guidelines.

However, these reported indoor FA concentrations are still varied among the studies. This discrepancy could be due to differences in some characteristics such as dimensions of dissection halls or rooms, types of forced ventilation [23] and number of cadavers in the dissection rooms [21]. Recently, a study in Germany reported that using both improved ventilation and a polymerized FA method in dissection rooms kept the indoor air concentration below the critical values [34]. Table 2 shows the comparison of indoor FA concentrations among different studies with respect to some characteristics of the dissection rooms. It is noted that, although the ventilatory system in our study may not be as effective as the type of ventilation utilized in other countries, the lowest number of cadavers and relatively greater dimension of the dissection rooms could favor the finding of comparable indoor FA concentrations.

It has been reported that the FA levels in dissection rooms can change between dissection sessions [23,36]. In our study, during the study period, 13 dissection sessions were investigated: four sessions on thorax and abdomen; five sessions on head, neck and brain; and four sessions on upper and lower limbs. The duration of these sessions ranged between 1 and 2 h, which in turn varied with regions of body dissected. Figure 6 shows the average indoor concentrations of FA calculated from four dissection rooms during each session. There was no statistically significant difference among the three portions of thorax and abdomen; head, neck and brain; and upper and lower limbs (*p* = 0.234). In the figure, it can be clearly seen that, during most of the dissection sessions, indoor FA concentrations fluctuated but were still above the guideline values set for occupational exposure. This fluctuation in indoor FA concentrations could be attributable to variations in regions of cadavers being dissected [20] and time spent on dissection [21,35].

### 3.2. Personal Exposure Levels of Formaldehyde

For all dissection rooms, the average value of personal FA exposure of the instructors was found to be higher than those of the students (Figure 7), and, in DR I, the mean personal exposure of the instructors was significantly higher than those of the students (*p* = 0.003). This finding is in agreement with the findings reported by some earlier studies [20,21,22]. For either demonstration of dissection or inspection of students’ dissection, instructors have to stand nearer to the cadavers than the students and this could partly be responsible for such a uniform report. In our study, sampling of personal exposure was done simultaneously to the instructors and the students as soon as the dissection sessions began. In fact, most instructors make the dissection about 0.5–1 h prior to the dissection session and, consequently, the instructors could have higher FA exposure than we observed.

Long-term exposure values in indoor guidelines are usually based on the 8-h time duration of a work shift. These time-weighted average (TWA) values are set to protect the individuals from the chronic effects of FA. In our study, the TWA values the individual exposed subjects to FA were calculated as TWA = $\frac{t\;\times \;c\;}{8\;hr}$, where t = 1.5 h (average duration of dissection sessions) and c is the individual’s FA exposure value. After analyzing TWA values of the exposed subjects, a positive skewness above 1.5 was observed for both exposed students and exposed instructors due to the fact that some of them showed extremely high personal FA exposures (Figure 8). This wide range may be attributable to the individual variations in the time spent in the dissection rooms [35] and in the distance between the cadavers and the exposed subjects [16].

The Occupational Safety and Health Administration has set the permissible exposure limit time-weighted average (PEL-TWA) at 0.75 ppm [37]. No individual in our study had a personal FA exposure above this set value. However, when compared with the guideline value, 0.1 ppm with 8-h TWA, for indoor air quality of office buildings set by Singapore Ministry of Environment [38] and Ministry of Labor in Korea [39], 33 out of 72 samplings of the exposed instructors (59.1%) and 23 out of 93 samplings of the exposed students (24.2%) showed personal FA exposure above this indoor workplace permissible limit.

Some previous studies which evaluated both indoor concentrations and personal exposure of FA in anatomy dissection rooms demonstrated that the personal concentrations were higher and varied more widely than their respective area concentrations [16,20,22,32], and a similar finding was also observed in our study (Table 3). The source of FA evaporation, i.e., the cadavers, are closer to personal exposure sampling than indoor air sampling, and hence such results were expected. At the same time, personal exposure being higher than indoor concentration indicates that the exposure to FA could be underestimated by using only data of area sampling [22]. However, data obtained from indoor air sampling are still helpful for assessment of indoor air quality of the dissection rooms.

A wider range of concentrations in personal exposure than indoor concentration could be due to the fact that area samplers are set at a fixed distance from the cadavers, whereas personal samplers are carried along with subjects who might be either close to or keep a distance from the cadavers. One previous study reported that, when a person is close to the cadaver, his/her personal exposure level was 2–3 times higher than mean indoor concentration [16].

An instructor or student represents a heat source; the approximately 100 Watts generated from each person and the increased number of persons surrounding the cadaver could increase the air turbulence, which in turn could increase the FA exposure level [4]. From the available data on the number of students per cadaver, in an Indian study on the dissection laboratory with 12 students per cadaver, the maximum personal FA exposure was 0.7 ppm [33]; in a Saudi Arabia study on the dissection room with 20 students per cadaver, the highest personal FA exposure of students was 1.42 ppm and that of instructors was 1.72 ppm; and, in our study, one cadaver was allotted to 25–30 students and the peak personal FA exposure reached up to 1.72 ppm in the students and 2.11 ppm in the instructors (Table 3). On the other hand, at the University of Occupational and Environmental Health, Kitakyushu, Japan, the maximum personal FA exposure of was only 0.06 ppm when five students were assigned to one cadaver (unpublished data).

### 3.3. Formaldehyde-Related Clinical Symptoms

Some individuals were highly sensitive to the FA while others were more resistant and had no reaction to the same level of exposure [40]. In our study, the most frequent complaint reported by the exposed students was the unpleasant smell (78.8%) (Figure 9), and this occurrence is consistent with some previous studies which also reported this symptom as the most predominant one at 68% [21], 80% [41], 77% [42], 57.5% [35], 91.2% [43] and, 70.3% [44]. This common finding simply reflects that FA exposure is mainly by inhalation during the dissection sessions.

Since irritation is one of the characteristics of FA [12], symptoms of eye and upper respiratory tract irritations such as eye itchiness, excessive lacrimation, eye redness, blurred vision and nasal symptoms were frequently observed in our study and other previous studies [21,35,42,43,45].

An interesting finding in our study was that 25.0% of the students experienced an inability to concentrate during the dissection sessions (Figure 9). Such complaint was also detected in some earlier studies which used questionnaires including symptoms such as lack of concentration, low assimilation or learning impairment during Anatomy practical classes at 23% [41], 32.1% [45] and 13.5% [43]. Moreover, a recent study also stated that 50% of the participating students complained of low assimilation during dissections [46]. This loss of concentration symptom could be related to general fatigue, headache and dizziness [41]. This finding is worth considering by the educational system because it can impair students’ learning progress [46]. In a study from Thailand, however, the most common clinical symptom was general fatigue (87.8%) [32], which was less frequently reported in other studies at 38.5% [36], 36% [21] and 21.8% [44], as well as 9.4% in our study.

In this study, we also analyzed any relationship of FA exposure to these subjective symptoms. It was found that, during the dissection sessions, the medical students were at a significantly higher risk of having these symptoms when compared to when they were in the lecture theater. Table 4 shows clinical symptoms which are significantly associated with FA exposure among the medical students. The result points out that these subjective symptoms could be specified to the exposure of FA encountered during dissection classes. This finding also conforms to some preceding studies, although differences in presenting symptoms exist between surveys. Significant differences in symptoms of burning eyes, itchy eyes, unusual thirst, bad feeling and fatigue were observed between the FA-exposed period and the non-exposed periods [23]. Mean scores of clinical symptoms related to FA exposure such as eye irritation, nose irritation, headache, itchy skin, shortness of breath, dry throat and chest tightness during working hours were significantly higher than during non-working hours [45].

Only 45% of the medical students used surgical masks regularly (Table 5), which could lead to a high percentage experiencing the unpleasant odor. Wearing activated carbon masks could decrease clinical symptoms induced by FA exposure in medical students [46]. Medical students who did not wear masks were adversely affected by certain clinical symptoms such as mouth dryness and respiratory distress [47].

No students in our study used goggles but about 40% of students reported that they wore eyeglasses assuming that eyes were partially protected from FA vapor. However, complaints of eye itchiness, excessive lacrimation and eye redness were still common among the students. The accumulation of FA vapors under the glasses could cause eye problems [48]. The medical students who wore glasses complained of excessive lacrimation and redness or itchiness of eyes [47]. Therefore, it could be supposed that not eyeglasses but only goggles could hamper the irritation effects of FA on eyes.

A study in Nigeria identified skin-related diseases as the least ranked effect of FA and gave a possible reason for protective wear in the hands and other parts of the body [42]. In agreement with this study, we also observed the itchiness of skin as the least frequent symptoms (2.3%), which might be because over 90% of the students regularly used laboratory coats and surgical gloves.

## 4. Conclusions

The indoor FA concentrations of the dissection rooms at the Department of Anatomy, UM1, were above the recommended values in terms of either short-term exposure limit for general population or occupational exposure limit. It is obvious that indoor air quality of the dissection rooms due to FA is undesirable. Personal FA exposure levels were found higher and varied more widely than indoor FA concentrations. The instructors suffered higher FA exposure than the students, while 59.1% of the instructors and about 24.2% of the students encountered FA exposure above the permissible workplace exposure limit. The medical students in our study were found to be at a significantly higher risk of having clinical symptoms of acute FA exposure during the dissection sessions than during lectures.

Although the ventilatory system was not as effective as the systems utilized in some countries, the lower number of cadavers and relatively greater dimensions of the dissection rooms in our study could favor the finding of similar indoor FA concentrations to those of other countries. On the other hand, greater number of allotted students per cadaver in the dissection rooms could lead to extremely high exposure among some exposed students and instructors. Common complaints of unpleasant odor and nose and eye irritations could be a consequence of failure in regular use of surgical mask and goggles.

This is the first study in Myanmar to assess the indoor FA level and personal exposure level of FA. More efficient ventilatory system, proper use of personal protective equipment, alternative use of low formaldehyde embalming techniques and reducing the number of students per cadaver should be considered for effective reduction in indoor concentration and personal exposure of FA. In addition, for the instructors working at the Department of Anatomy, a longitudinal study is required to investigate any risk of diseases such as hematological problems and cancers which have been recognized during chronic exposure to formaldehyde.

## Figures and Tables

**Figure 1 ijerph-18-00712-f001:**
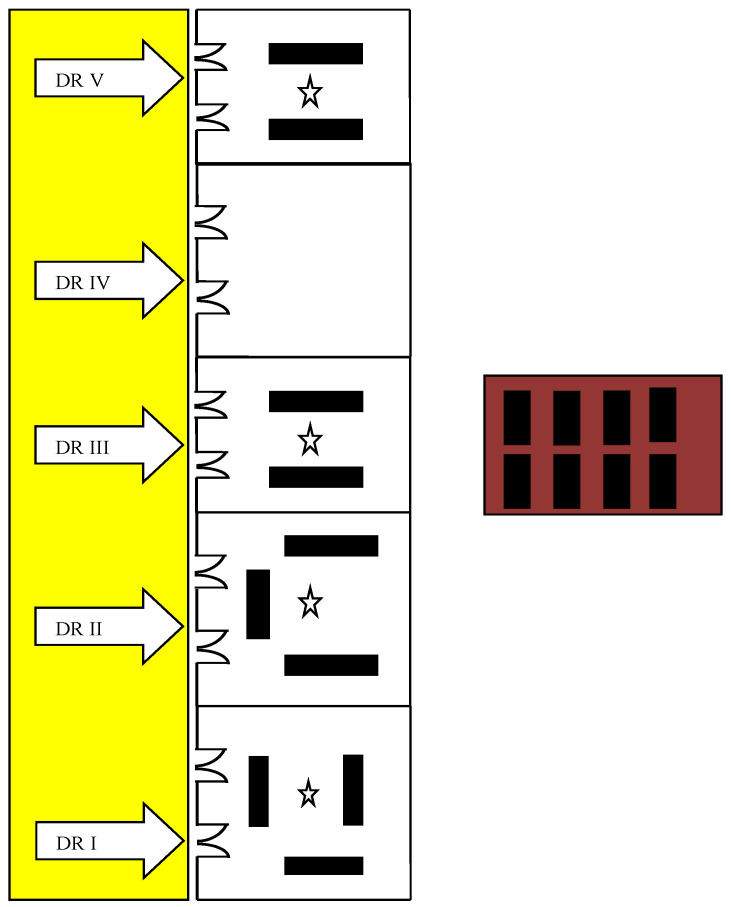
Description of study location (
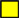
 Corridor, 
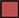
 Cadaver storage room, 
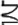
 Door (DR, dissection room), 
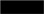
 Cadaver, and 
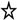
 Diffusive air sampler).

**Figure 2 ijerph-18-00712-f002:**
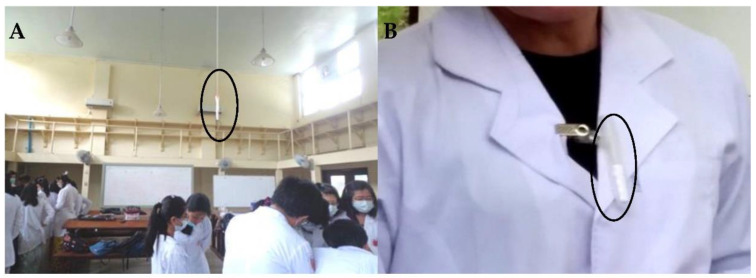
Diffusive air samplers: (**A**) air samplers placed in dissecting room for indoor FA concentration; and (**B**) attached to the breathing zone for personal FA exposure.

**Figure 3 ijerph-18-00712-f003:**
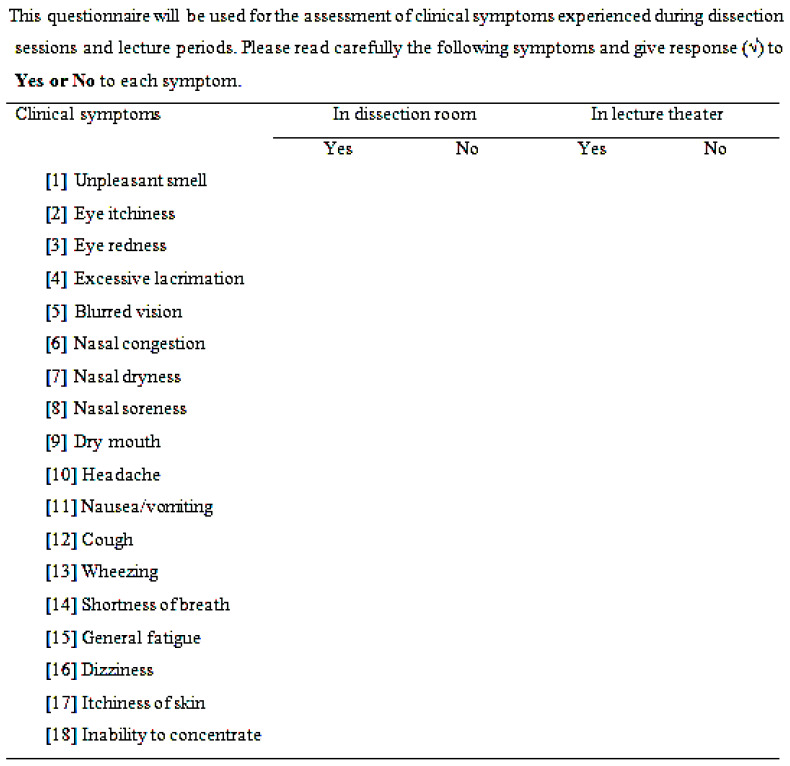
Questionnaire for the assessment of formaldehyde-related clinical symptoms.

**Figure 4 ijerph-18-00712-f004:**
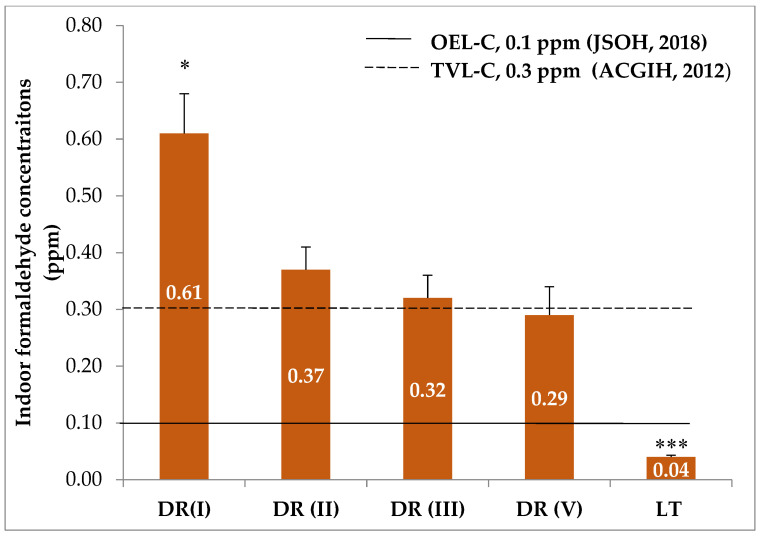
Indoor formaldehyde concentrations in dissection rooms and lecture theater compared with guideline values (mean ± SEM). * *p* < 0.05, indoor concentration in DR I significantly higher than those in other rooms; *** *p* < 0.001, indoor concentration in LT significantly lower than those in DRs. OEL-C, occupational exposure limit-ceiling; TVL-C, threshold time value-ceiling; DR, dissection room; LT, lecture theater.

**Figure 5 ijerph-18-00712-f005:**
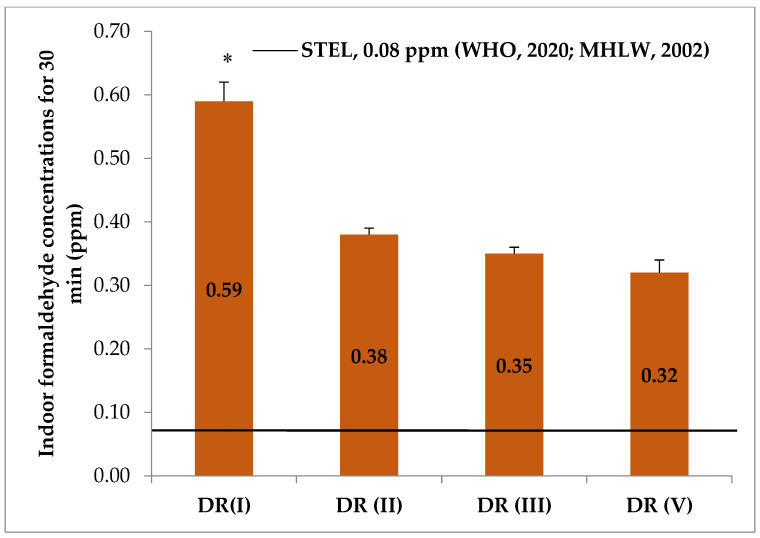
Indoor formaldehyde concentrations in dissection rooms and lecturer compared with short term exposure limit for 30 min (mean ± SEM). * *p* < 0.05, indoor concentration in DR I significantly higher than those of other rooms. STEL, short-term exposure limit; DR, dissection room; LT, lecture theater.

**Figure 6 ijerph-18-00712-f006:**
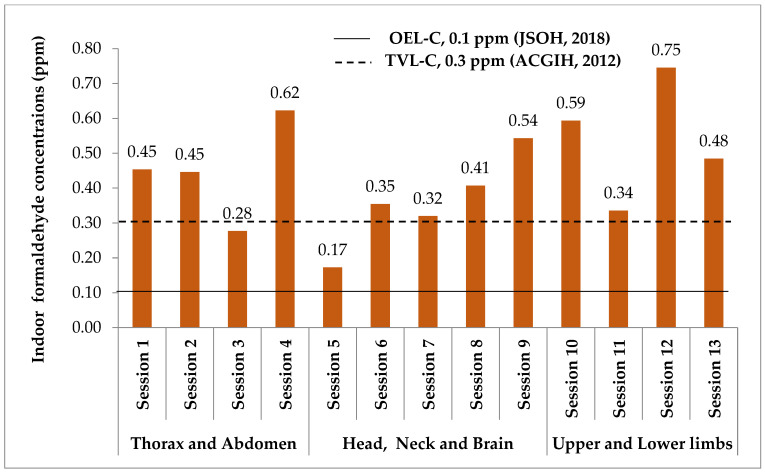
Indoor formaldehyde concentrations in dissection rooms during each dissection session throughout the study period each bar represents the mean of four dissection rooms). OEL-C, occupational exposure limit-ceiling; TVL-C, threshold time value-ceiling.

**Figure 7 ijerph-18-00712-f007:**
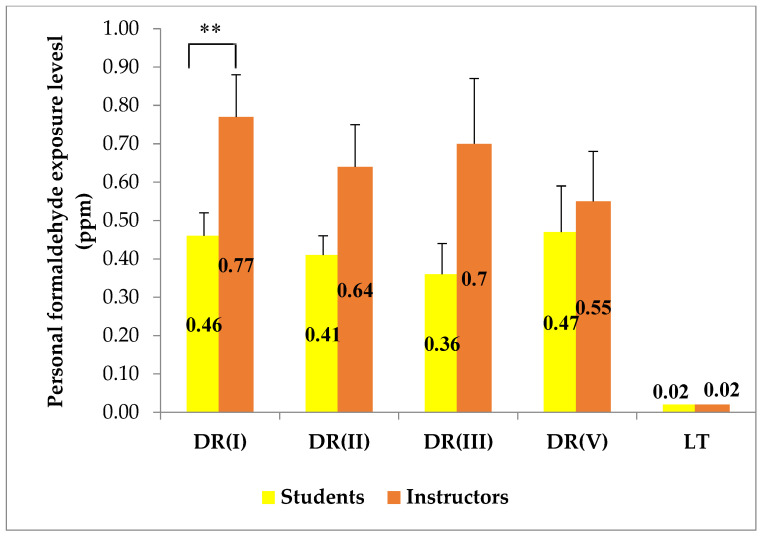
Personal exposure levels of instructors/ lecturers and students in the dissection rooms and lecture theater (mean ± SEM). ** *p* < 0.01. DR, dissection room; LT, lecture theater.

**Figure 8 ijerph-18-00712-f008:**
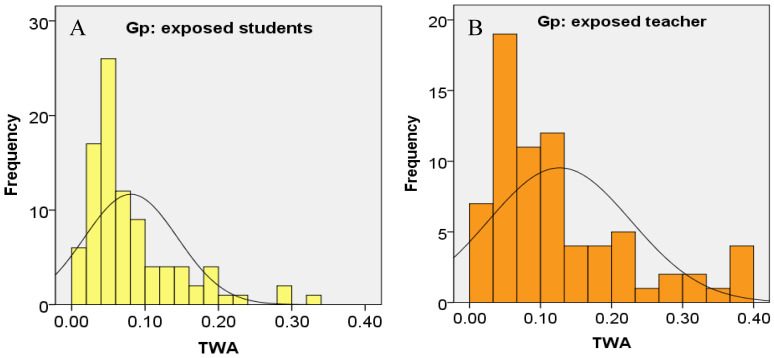
Frequency distribution of personal FA exposure in term of time-weight average in: (**A**) exposed students; and (**B**) exposed instructors.

**Figure 9 ijerph-18-00712-f009:**
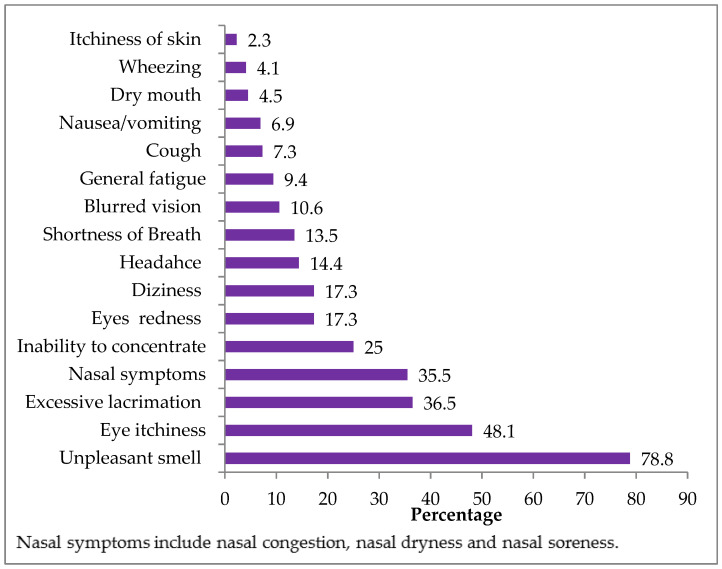
Percentage of FA-related clinical symptoms reported by exposed subjects (*n* = 104).

**Table 1 ijerph-18-00712-t001:** Characteristics of dissection rooms and lecture theater.

Characteristics	DR(I)	DR(II)	DR(III)	DR(V)	LT
(1) Dimension (m^3^)	1188	1188	924	924	1597
(2) Type of forced ventilation	Wall-mounted Fans	Wall-mounted Fans	Wall-mounted Fans	Wall-mounted Fans	Air conditioning
(3) Number of cadavers	3	3	2	2	Nil
(4) Temperature (°C)	27–28	27–28	27–28	27–28	22–24
(5) Relative Humidity (%)	70–85	70–85	70–85	70–85	50–65

**Table 2 ijerph-18-00712-t002:** Comparison of indoor FA concentrations among different studies with regard to characteristics of dissection rooms.

Authors and Year	Dimension (m^3^)	Type of Ventilation	Cadaver Number	Indoor FA Concentrations (ppm)
(1) Ohmichi et al. (2006) [16]	1125	12 supply diffuser, 4 air conditioning, 8 air return grills	not mention	0.23–1.03
(2) Vohr et al. (2011) [20]	1800	Central air conditioning system, 4 exhaust fans	14	0.68 (Week 4) 0.85 (Week 10) 0.73 (Week 14)
(3) Kunugita et al. (2004) [23]	1400	General ventilation (2001)	25	0.76
		General ventilation with new filter (2020)		0.61
		Novel local ventilatory system for each dissection table (2004)		0.05
(4) Azari et al. (2012) [21]	339	No ventilatory system, Exhaust Off		0.57
		Only ventilatory supply system		0.27
		Ventilatory supply system, Exhaust On		0.25
(5) Gahukar et al. (2014) [33]	1386	Natural ventilation system	16	
(6) Homwutthiway & Ongwandee (2017) [35]	380	4 large stand fans	14	0.09–0.176
(7) The present study (2020)	11,188 (DRs I, II)	10 wall fans	3 (DRs I, II)	DR I, 0.61 DR II, 0.37
	9942 (DRs III, V)		2 (DRs III, V)	DR III, 0.32 DR V, 0.29

For proper comparison, concentrations in μgm^−3^ or mgm^−3^ in some original articles were converted into ppm.

**Table 3 ijerph-18-00712-t003:** Comparison of indoor FA concentrations and personal exposure among different studies with regard to characteristics of dissection rooms.

Authors and Year	Country	Concentration of FA (ppm) Min-Max or Average (Min-Max)	
		Indoor Concentration	Personal Exposure
(1) Ohmichi et al. (2006) [16]	Japan	0.24–1.05	0.46–1.1
(2) Vohra (2011) [20]	Saudi Arabia	4th week (Upper limb)	1.27 (0.82–1.72) ^I^
		0.68 (0.53–0.80)	0.75 (0.62–0.89) ^S^
		10th week (Abdomen)	1.44 (1.18–1.70) ^I^
		0.85 (0.64–1.29)	1.20 (0.98–1.42) ^S^
		14th week (Head, Neck)	1.33 (0.94–1.72) ^I^
		0.73 (0.65–0.93)	1.10 (0.88–1.31) ^S^
(3) Ochs et al. (2012) [22]	Brazil	0.15–1.89	0.41–2.80
(4) Lakchayapakorn and Watchalayarn (2010) [32]	Thailand	0.49 (0.40–0.58)	0.66 (0.47–0.85)
(5) Present study (2020)	Myanmar	0.43 (0.09–1.22)	0.68 (0.04–2.11) ^I^ 0.44 (0.06–1.72) ^S^

^I^ Instructor, ^S^ Student.

**Table 4 ijerph-18-00712-t004:** Relationship of formaldehyde exposure to the clinical symptoms, based on the analysis of paired samples (n = 104 students).

Clinical Symptoms	Number of Students Who Reported Symptoms (%)		Odd Ratio (95% CI)
	In Dissection Room	In Lecture Theater	
(1) Unpleasant smell	82 (78.8%)	5 (4.8%)	39.4 (17.2–90.2) ***
(2) Eye itchiness	50 (48.1%)	10 (9.6%)	22.3 (7.6–65.0) ***
(3) Excessive lacrimation	38 (36.5%)	3 (2.9%)	19.3 (5.7–65.4) ***
(4) Nasal symptoms	37(35.5%)	10 (9.6%)	5.2 (2.4–11.2) ***
(5) Inability to concentrate	26 (25.0%)	5 (4.8%)	5.4 (2.1–13.9) ***
(6) Eye redness	18 (17.3%)	3 (2.9%)	7.0 (2.0–24.7) **
(7) Dizziness	18 (17.3%)	4 (3.8%)	4.2 (1.3–13.2) **
(8) Headache	15 (14.4%)	6 (5.8%)	2.8 (1.0–7.4) *
(9) Shortness of breath	14 (13.5%)	5 (4.8%)	3.1 (1.1–8.9) *
(10) Blurred vision	11 (10.6%)	3 (2.9%)	3.9 (1.1–14.7) *

Analyzed by simple logistic regression. *** *p* < 0.001, ** *p* < 0.01, * *p* < 0.05.

**Table 5 ijerph-18-00712-t005:** Percentage of respondents reporting regular use of personal protective equipment during dissection sessions.

Regular Use of PPE	Percentage of Respondents (n = 104)
(1) Laboratory coat	96%
(2) Surgical gloves	91%
(3) Surgical masks	45%
(4) Goggles	0%

## Data Availability

Data are available on request to corresponding author.

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
