# Peer review of "Indoor Formaldehyde Concentration, Personal Formaldehyde Exposure and Clinical Symptoms during Anatomy Dissection Sessions, University of Medicine 1, Yangon"

_ijerph, 2021, doi:10.3390/ijerph18020712_

Round 1

Reviewer 1 Report

Interesting topic, good study and manuscript, I recommend just minor revision of the text before publication 

Please find below some comments and suggestions to the authors of this manuscript.   1. Since the chronic FA exposure of the instructors looks higher, due to their professional duties, it would be interesting to follow the 18 members of the teaching staff for longer time, in terms of general clinical aspects and main risks, such as hematological and neoplastic diseases.    2. Some comparative data referring to FA exposure, alternative substances and protective measures used in other big medical universities from Europe and North America might be useful.   3. Two small wording errors :  - line 22- of medical students ... - Figure 6-...xposure  levels ...

Author Response

REVIEWER#1

Interesting topic, good study and manuscript, I recommend just minor revision of the text before publication 

  • It is very grateful for such commend and we really appreciate it.

Please find below some comments and suggestions to the authors of this manuscript.  

  1. Since the chronic FA exposure of the instructors looks higher, due to their professional duties, it would be interesting to follow the 18 members of the teaching staff for longer time, in terms of general clinical aspects and main risks, such as hematological and neoplastic diseases.   
  • Thank you very much for your kind suggestions. We have a plan to perform a longitudinal study in order to investigate any risk of some diseases like hematological problems and cancers in these instructors having chronic exposure to formaldehyde.
  • Please kindly check in the section 4, the line number 430-433 in the conclusion section.
  1. Some comparative data referring to FA exposure, alternative substances and protective measuresused in other big medical universities from Europe and North America might be useful.  
  • According to the Reviewer’s advice, we cited a reference from Germany in which the authors determined the effects of low cost and efficient reduction of formaldehyde concentration in gross anatomy rooms. Please kindly see in the section 1, the line number 262-264.  
  1. Two small wording errors :  - line 22- of medical students ... - Figure 6-...xposure  levels ...

Thank you for checking such wording errors. The errors were corrected accordingly. Please kindly see line number- 23 and 307.

Reviewer 2 Report

This manuscript pretends to evaluate indoor formaldehyde concentration, personal formaldehyde exposure and clinical symptoms for both students and instructors during Anatomy dissection sessions.

The article is well structured and easy to read.

Some specific comments/suggestions:

Line 21: I suggest to refer that the lecture theater is the control group.

Line 127: designation of group A and B only appears in this line…

Line 170:  “…104 medical students were randomly selected after excluding those with a history of allergy or skin disorders.” How were they selected? Some from all the DR and the same number from the LT? Are 104 for each group? This part is cofuse in my opinion.

Line 190: “Statistically significant level was determined at p < 0.05” Not always was this the value considered, see for example line 205. I suggest to always refer the p-value obtained for each situation and not just state that was or not significant at a given level. Also refer if the assumptions for the applied statistic tests were validated.

Line 210: In my opinion this should not be in the figure legend (at least not only here).  

Line 212: “Although DR I vs. DR II and DR III vs. DR V have the same dimension and the same number of cadavers respectively (Table 1), each two rooms with such similarities still showed different indoor FA concentrations…”. This isn´t the result of the statistic test… For example, results of DR III and DR V are different or not?

Line 229 : same as line 210.

Line 258: Why not making a statistical test to see if there are significant differences?

Line 266: Add statistically and refer p values.

Line 360: First symptom in table 4, shouldn´t it be 78.8% instead of 80.8%? How is the odd ratio determined? Is it the ratio between the odd (DR) and the odd(LT)? This is not clear for me, could you please explain?

Author Response

REVIEWER#2

This manuscript pretends to evaluate indoor formaldehyde concentration, personal formaldehyde exposure and clinical symptoms for both students and instructors during Anatomy dissection sessions.

The article is well structured and easy to read.

  • Thank you very much for your comment to our manuscript.

Some specific comments/suggestions:

  1. Line 21: I suggest to refer that the lecture theater is the control group.
  • Yes, as your suggestion, we added the words “an exposed site” behind “Anatomy dissection rooms” “and an unexposed site” behind “lecture theater” so as to clarify between these two sites. Please check the line number 21-22.

  1. Line 127: designation of group A and B only appears in this line…
  • Such defining group was just for an example. Actually, we wanted to say that the second-year medical students (1/2019 batch) was divided to two groups namely, group A and group B. Neither group A nor group B was consistently the exposed group in the dissection rooms or unexposed group in lecture theater for the whole academic year. According to the time-table, group A or group B could be the exposed group alternatively. Therefore, not to be confused, we add some new sentences for better understanding. Please kindly check the line number 131-134.

  1. Line 170:  “…104 medical students were randomly selected after excluding those with a history of allergy or skin disorders.” How were they selected? Some from all the DR and the same number from the LT? Are 104 for each group? This part is confuse in my opinion.
  • We are sorry for making confusion to the Reviewer/ readers. The assessment of FA-related symptoms by the questionnaire done on a day just before the end of the study period for the reason to avoid the effects of fluctuations in indoor FA concentrations between the dissection sessions. On that day, the students were first explained about the exclusion criteria; any history of allergy and skin disorders and then asked for voluntary participation to answer the questionnaire. Then, the students were randomly selected by lottery method in which who received a tick mark would have to fill the questionnaire and who got no mark would not have to answer. The students who filled and returned the questionnaire included from both group A and group B.
  • Actually, although the sites were defined as exposed (the dissection rooms) and unexposed (lecture theater) sites, all the students had already experienced to FA exposure during their respective dissection sessions because this assessment was done on a day just before the end of the study period.
  • For better understanding regarding the procedure of assessment of FA-related symptoms, we revised the section 6Questionnaire for assessment of formaldehyde-related clinical symptoms”. Please check the line number 180, 187-189, 191-195. A figure showing the questionnaire is also added and this figure number is 3 and the following figure numbers change accordingly.

  1. Line 190: “Statistically significant level was determined at p < 0.05” Not always was this the value considered, see for example line 205. I suggest to always refer the p-value obtained for each situation and not just state that was or not significant at a given level. Also refer if the assumptions for the applied statistic tests were validated.
  • Thank you very much for your expert suggestion for statistical analysis. According to your suggestion, we added the actual p-values to the all statements expressing the comparison of the numerical variables such as indoor FA concentrations among four dissection rooms and lecture theater and personal FA exposure levels between the instructors and the students.
  • Please kindly check the line number- 221, 230-233, 278-280, and 290-291.
  • For comparison of indoor the formaldehyde concentrations among the dissection rooms and lecture theater, the. following assumptions were validated for the applied statistic test, ANOVA
  • The samples were drawn randomly from the population of interest.
  • The groups were unrelated and independent samples.
  • The variable of interest was normally distributed in each group.
  • Homogeneity of variance: the groups came from the population with equal variances and Levene’s test was used to check for equality of variance. Since Levene’s test showed P<0.05, equal variance was assumed and Post Hoc “Bonferroni” was used for multiple comparisons.

  • For comparison of personal exposure levels between the instructors and the students, the following assumptions were validated for the applied statistics test, Independent sample “t “test
  • The samples were drawn randomly from the population of interest.
  • Independent sample: the sample appeared in only one group and these two groups were unrelated.
  • The variable of interest was normally distributed in each population.
  • Homogeneity of variance: the groups should come from the population with equal variance and Levene’s test was used to check for equality of variance.
  • For assessment of risk of having FA-related symptoms in FA-exposed students, odd ratio was calculated by using Simple Logistic Regression and the following assumptions were checked for such analysis.
  • The two conditions, exposed condition in the dissection rooms and unexposed condition in the lecture theater did not affect each other.
  • There was no linear relationship between the dependent and independent variables.
  • There was a linear relationship between the independent variables and log odds.
  1. Line 210: In my opinion this should not be in the figure legend (at least not only here).
  • According to your advice, we delete the name of statistical tests in all the figure legends. Please check the line number 225, 248-249, 308.

  1. Line 212: “Although DR I vs. DR II and DR III vs. DR V have the same dimension and the same number of cadavers respectively (Table 1), each two rooms with such similarities still showed different indoor FA concentrations…”. This isn´t the result of the statistic test… For example, results of DR III and DR V are different or not?
  • Yes, there was a significant difference in indoor FA concentrations between DRI and the remaining three rooms. No statistically significant difference in indoor FA concentrations was seen among DRII, DR III and DR V.
  • We answer the Reviewer’s questions on any statistical difference between the dissection rooms and we add some sentences at line number 231-234.
  •  
  1. Line 229 : same as line 210.
  • Please check the line number 248-249.

  1. Line 258: Why not making a statistical test to see if there are significant differences? Line 266: Add statistically and refer p values.
  • Since we mainly focused on the situation how much indoor FA concentrations fluctuated among the dissection sessions and only mean value (from 4 dissection rooms) was expressed for each dissection session, statistical analysis of indoor FA concentrations among 13 sessions was not done. However, we accept your suggestion as much as possible we can; we performed statistical analysis among three main portions; thorax and abdomen, head, neck and brain and upper and lower limbs. The SPSS output showed that no statistically significant difference among these portions, p=0.234. We added this information of statistical analysis at the line number 277-279.

  1. Line 360: First symptom in table 4, shouldn´t it be 78.8% instead of 80.8%? How is the odd ratio determined? Is it the ratio between the odd (DR) and the odd(LT)? This is not clear for me, could you please explain?
  • Yes, as you pointed-out, the percentage of the first symptom, unpleasant smell should be 78.8%. Accordingly, we correct this percentage at the Figure 9 and the Table 4.

For the questionnaire, the response was Yes or No response to the symptoms regarding two different environments- in the dissection rooms and in the lecture theater. Based on the analysis of such paired-samples, we calculated Odd Ratio or risk of having FA-related symptoms in the dissection rooms by using Simple Logistic Regression (SLR).We defined the unexposed site (the lecture theater) as reference (0) and exposed site (the dissection room) as 1. And, from the SPSS output of SLR, we took the value of exponential function of the slope, Exp(B) as the value of OR.

Reviewer 3 Report

In the present manuscript the Authors present the results of a study performed on medical students and aimed at evaluating their indoor exposure to formaldehyde during anatomy dissection sessions. The main finding of the study is that exposure of Students and Instructors to formaldehyde easily exceeds the limits for individual exposure thus causing formaldehyde-related symptoms.

This is a potentially interesting study, but it is difficult to read due to a complex methodological design poorly presented and with too many comparisons not clearly explained: Students vs Instructors, Frequency distribution of personal FA exposure for Students vs Instructors, Dissection rooms vs Lecture theater, Dissection rooms among them overall and for 30 minutes, Dissection type, Symptoms.

Many tables and figures in which the legends are often not able to frame the context the numbers/graph refer to.

Specific concerns

Page 5, row 129: “13 dissection sessions were investigated” Per each room? In total?

Page 5, row 139-140: “Six diffusive air 139 samplers were utilized for one cross-sectional measurement of indoor FA concentrations” What is this aiming for?

Page 144-155: please, let mi understand how DSD-BPE/DNPH samplers work. Is their result time-dependent (e.g., as for Radiello® monitors)? Is so, no mention is made of any record of their exposure time.

Page 6, row 169, paragraph 2.6: There is no mention of the questionnaires submitted to students in Lecture theater

Page 6, row 184, paragraph 2.7 “Statistical analysis”: ANOVA and t-test work for normally distributed continuous variable. Please, declare that – and how – you checked the normality of data distribution.

Page 7, Figure 3 and all the following: Figure legends have to be self-explaining. Labes as OEL-C, TVL-C, DR, LT, are nor explained. Moreover, I cannot see anywhere the number of subjects each column is related to. Why do you use SEM instead of SD?

Page 8, Figure 4: The measured FA levels shown in Fig. 4 are very similar to those in Fig. 3. Because they are measured in the same conditions with the only difference – if I well understand – of the duration of measurement, I think that Fig. 4 is not necessary

Page 8, row 241: “assisted ventilation” is usually adopted for mechanical ventilation in critically ill patients, probably, for buildings, better "forced"

Page 10, Figure 5: is each bar related to a single measurement? If no, bar deviation are missing.

Page 13, row 350: In Methods section, no mention is made of the evaluation performed by questionnaires on (how many? again 104?) unexposed students.

Page 14, row 360-361: Table 4 legend. Please explain the model on which this analysis is based. What do you mean when you say “paired samples”? You are not performing a paired comparison but a comparison between two independent sample (probably) composed of 104 subjects each.

Page 14, Table 4. It mysterious for me. When looking at “In lecture theater”, I can see that 5 is 4.8% and 10 is 2.5%. Because I have to suppose that the denominator (104?) is always the same, this cannot be correct! Look at “Eye itchiness” row: I tried to compute the raw OR for Eye itchness by considering 50 subjects reporting symptoms (48.1% of 104) in Dissection room and, consequently, 54 not reporting. In Lecture theater: 10 subjects reporting symptoms (10 is 2.5% of 400!) and consequently 390 not reporting. The raw OR was 36.1 (17.3 - 75.4). What did I do wrong?

Author Response

REVIEWER#3

In the present manuscript the Authors present the results of a study performed on medical students and aimed at evaluating their indoor exposure to formaldehyde during anatomy dissection sessions. The main finding of the study is that exposure of Students and Instructors to formaldehyde easily exceeds the limits for individual exposure thus causing formaldehyde-related symptoms.

This is a potentially interesting study, but it is difficult to read due to a complex methodological design poorly presented and with too many comparisons not clearly explained: Students vs Instructors, Frequency distribution of personal FA exposure for Students vs Instructors, Dissection rooms vs Lecture theater, Dissection rooms among them overall and for 30 minutes, Dissection type, Symptoms.

Many tables and figures in which the legends are often not able to frame the context the numbers/graph refer to.

Specific concerns

  1. Page 5, row 129: “13 dissection sessions were investigated” Per each room? In total?
  • We investigated a total of 13 dissection sessions within the study period and indoor FA concentrations were assessed in each dissection room. And we described the fluctuations of indoor FA concentrations among 13 dissection sessions in Figure 6 and each bar represented mean value of indoor concentration from four dissection rooms.

  1. Page 5, row 139-140: “Six diffusive air 139 samplers were utilized for one cross-sectional measurement of indoor FA concentrations” What is this aiming for?
  • Yes, as we mentioned above, we assessed indoor FA concentrations in the dissection rooms and lecture theater simultaneously. There were four dissection rooms (DRI, DR II, DRIII and DR V) and we used one air sampler in each dissection room and for the control, in the large lecture theater, one air sampler at the frontline and one sampler at the backline. Therefore, for one dissection session, a total of six air samplers were used for one cross-sectional measurement of indoor FA concentrations. In this way, we assessed 13 dissection sessions during the study period.
  1. Page 144-155: please, let me understand how DSD-BPE/DNPH samplers work. Is their result time-dependent (e.g., as for Radiello® monitors)? Is so, no mention is made of any record of their exposure time.
  • The diffusive sampling device, DSD-DNPH, is a porous sintered polyethylene tube as a permeable diffusion barrier and DNPH (2,4-dinitrophenylhydrazine) is a reaction absorbent loaded onto silica gel particles.
  • The device comprises of three sections;
  1. PSP-diffusion filter ; a porous sintered polyethylene
  2. PP-reservoir : polypropylene tubing
  • DNPH coated silica gel :250 mg of silica gel (105-210μm) coated with 1.2 mg of DNPH containing phosphoric acid
  • Please kindly check the figure (ref. Uchiyama et al., 2004)
  • And the sampling rate is fairly constant between 30 to 240 minutes and average sampling rate is 71.9 ml/min. Therefore, the sampling rate and capacity of the sampler is not time-dependent. We add this information at the line number 142-144.
  • Occupational Safety and Health Administration (OSHA) tested the capacity and sampling rates of DSD-DNPH (https://www.osha.gov/dts/sltc/methods/studies/srvsupelco/srvsupelco.html).

    Please kindly check the following table.   

Time (min)    Sampling rate and Capacity (ml/min)

5                    67.66

10                  68.32

15                  68.88

30                  70.32

60                  71.24

120                71.59

180                70.78

240                68.65

360                62.18

480                52.54

  1. Page 6, row 169, paragraph 2.6: There is no mention of the questionnaires submitted to students in Lecture theater
  • We are sorry for making confusion to the Reviewer/ readers. The assessment of FA-related symptoms by the questionnaire was done on a day just before the end of the study period for the reason to avoid the effects of fluctuations in indoor FA concentrations between the dissection sessions. On that day, there was no dissection session and some revision lectures were given to both group A and group B.
  • The students were first explained about the exclusion criteria such as any history of allergy and skin disorders and asked for voluntary participation to answer the questionnaire. Then, the students were randomly selected by lottery method in which who received a tick mark would have to fill the questionnaire and who got no mark would not have to answer. The students who filled and returned the questionnaire included from both group A and group B.
  • The students were asked to fill the questionnaire by recalling whether they experienced the described symptoms either in the dissection room or the lecture theater.
  • Please kindly check the following figure that showed the questionnaire used in our study.
  • We would like to add this figure showing the questionnaire to the revised manuscript in the section 2 and this figure number is figure 3 and the following figure numbers will be changed accordingly.

  1. Page 6, row 184, paragraph 2.7 “Statistical analysis”: ANOVA and t-test work for normally distributed continuous variable. Please, declare that – and how – you checked the normality of data distribution.
  • The distributions of these continuous numerical variables were checked by two methods namely, shape of the histogram and the value of the skewness. The numerical variables checked for distribution showed the skewness <1 and so they were considered as normally distributed continuous variables.

  1. Page 7, Figure 3 and all the following: Figure legends have to be self-explaining. Labes as OEL-C, TVL-C, DR, LT, are nor explained. Moreover, I cannot see anywhere the number of subjects each column is related to. Why do you use SEM instead of SD?
  • Yes, according to the Reviewer’s kind suggestion, we will explain the initial term in the respective figure legends. Please kindly check the figures.
  • We attached the air samplers at the breathing zone of the instructors and the students and measured their personal FA exposure level. Figure 7 shows how much FA exposure was high in the instructors and the students and Figure 8 shows the frequency distribution of personal exposure levels among exposed instructors and students.
  • In Figure 8, for example, the highest frequency, 27 exposed students, showed TWA between 0.04-0.06 ppm.
  • We preferred SEM because it quantifies how far the estimation of the mean likely to be from the true population mean.

  1. Page 8, Figure 4: The measured FA levels shown in Fig. 4 are very similar to those in Fig. 3. Because they are measured in the same conditions with the only difference – if I well understand – of the duration of measurement, I think that Fig. 4 is not necessary
  • Yes, the Figure 4 (air sampling for complete duration of dissection sessions, 1-2 hr) and Figure 5 (air sampling for only 30 minutes) show the indoor FA concentrations with different duration of measurements.
  • We assumed that in order to recommend whether indoor FA concentration in the DRs was above or below the set values for any 30 minutes of acute exposure time set by WHO and MHLW, it would be more appropriate by comparing with the same duration of 30 minutes.

  1. Page 8, row 241: “assisted ventilation” is usually adopted for mechanical ventilation in critically ill patients, probably, for buildings, better "forced".
  • Thank you for your suggestion. We changed the word “assisted’ into “forced”. Please kindly check the line numbers- 80, 261: Table 1-No.2.

  1. Page 10, Figure 5: is each bar related to a single measurement? If no, bar deviation are missing.
  • each bar represents mean value of indoor FA concentrations measured at four dissection rooms at each dissection session. We mainly focused to describe the fluctuation in FA concentrations among dissection session, also wanted to showed comparison with the set values and SD could be high for only four samples i.e. four dissection rooms. For these three reasons, we decided to not to add bar deviation. We hope you will accept our explanation.

  1. Page 13, row 350: In Methods section, no mention is made of the evaluation performed by questionnaires on (how many? again 104?) unexposed students.
  • Actually, these 104 students were randomly selected from both group A and group B. All of them had been exposed to FA in the dissection sessions during the study period and hence they all were exposed students. They were asked to fill the questionnaire and gave response “Yes” or “No” to the listed FA-related clinical symptoms of which they experienced either during the dissection sessions or during lecture.

  1. Page 14, row 360-361: Table 4 legend. Please explain the model on which this analysis is based. What do you mean when you say “paired samples”? You are not performing a paired comparison but a comparison between two independent sample (probably) composed of 104 subjects each.
  • Please kindly check the figure mentioned earlier that showed the questionnaire used to assess FA-related symptoms. The subjects were only 104 who already had exposed to FA during the dissection sessions. By recalling how they felt during dissection sessions (exposed site) and lecture periods (unexposed site), they were asked to give Yes or No response to the given FA-related symptoms in these two different conditions and so it is the paired comparison and the samples are paired samples.
  • The comparison we made was not the comparison between two independent samples i.e. unexposed students and exposed students but comparison between the symptoms experiences during the dissection sessions and during lecture periods reported by the exposed students.

  1. Page 14, Table 4. It mysterious for me. When looking at “In lecture theater”, I can see that 5 is 4.8% and 10 is 2.5%. Because I have to suppose that the denominator (104?) is always the same, this cannot be correct! Look at “Eye itchiness” row: I tried to compute the raw OR for Eye itchness by considering 50 subjects reporting symptoms (48.1% of 104) in Dissection room and, consequently, 54 not reporting. In Lecture theater: 10 subjects reporting symptoms (10 is 2.5% of 400!) and consequently 390 not reporting. The raw OR was 36.1 (17.3 - 75.4). What did I do wrong?
  • First, we want to apologize for making gross mistake of typing error for some percentages. The denominator is always the same, 104, for both in the dissection room and in the lecture theater.
  • We corrected these wrong percentages regarding
  • % of students showing feeling of unpleasant smell in the dissection room- from 80.8% to 78.8%.
  • % of students suffering eye itchiness in the lecture theater from 2.5% to 9.6%
  • % of students showing nasal symptoms in the lecture in the lecture theater from 2.5% to 9.6.
  • For eye itchiness, 50 out of 104 students reported this symptom in the dissection rooms and 10 out of these 104 students reported this symptom in the lecture theater and the exponential function of the slope- exp(B) was 22.3 (95%CI, 7.6-65.0). According to the finding, we can say that the students were at 22.3 times higher risk of having eye itchiness in the dissection rooms than in the lecture theater.
  • We chose binary logistic regression for calculation of OR because there are less likely to make type I error when compared with Chi-square test.

Round 2

Reviewer 3 Report

The quality of manuscript improved after revision

Author Response

Responses to the Reviewers’ comments

Reviewer # 3

We did not find any suggestion and comments in ROUND 2.

Please let us know suggestion and comments and we will respond as soon as possible.

Thank you for your kind guidance.
